# Heat Capacities of l-Histidine, l-Phenylalanine, l-Proline, l-Tryptophan and l-Tyrosine

**DOI:** 10.3390/molecules26144298

**Published:** 2021-07-15

**Authors:** Václav Pokorný, Vojtěch Štejfa, Jakub Havlín, Květoslav Růžička, Michal Fulem

**Affiliations:** 1Department of Physical Chemistry, University of Chemistry and Technology Prague, Technická 5, 166 28 Prague 6, Czech Republic; pokornyv@vscht.cz (V.P.); stejfav@vscht.cz (V.Š.); ruzickak@vscht.cz (K.R.); 2Central Laboratories, University of Chemistry and Technology Prague, Technická 5, 166 28 Prague 6, Czech Republic; havlinj@vscht.cz

**Keywords:** l-histidine, l-phenylalanine, l-proline, l-tryptophan, l-tyrosine, crystal heat capacity, thermodynamic functions

## Abstract

In an effort to establish reliable thermodynamic data for proteinogenic amino acids, heat capacities for l-histidine (CAS RN: 71-00-1), l-phenylalanine (CAS RN: 63-91-2), l-proline (CAS RN: 147-85-3), l-tryptophan (CAS RN: 73-22-3), and l-tyrosine (CAS RN: 60-18-4) were measured over a wide temperature range. Prior to heat capacity measurements, thermogravimetric analysis was performed to determine the decomposition temperatures while X-ray powder diffraction (XRPD) and heat-flux differential scanning calorimetry (DSC) were used to identify the initial crystal structures and their possible transformations. Crystal heat capacities of all five amino acids were measured by Tian–Calvet calorimetry in the temperature interval from 262 to 358 K and by power compensation DSC in the temperature interval from 307 to 437 K. Experimental values determined in this work were then combined with the literature data obtained by adiabatic calorimetry. Low temperature heat capacities of l-histidine, for which no literature data were available, were determined in this work using the relaxation (heat pulse) calorimetry from 2 K. As a result, isobaric crystal heat capacities and standard thermodynamic functions up to 430 K for all five crystalline amino acids were developed.

## 1. Introduction

This work represents a continuation of our previous project [1,2,3,4], which aimed to establish reliable thermodynamic data along the saturation curve for proteinogenic amino acids. Despite their significance [5,6] and wide use, for example, in pharmaceutical formulations as coformers used to increase the bioavailability of poorly water soluble drugs [7,8], there are still considerable gaps in the availability of fundamental thermodynamic data for basic amino acids [2,9,10]. In addition, the values for properties such as sublimation or crystal-phase enthalpies of formation often show very poor interlaboratory agreement [2,10]. The discrepancies in sublimation thermodynamic properties can be ascribed to the zwitterionic nature of amino acid crystals, which causes their extremely low volatility and renders the experimental determination of their sublimation pressure and enthalpy difficult, prone to systematic errors, or even unfeasible at room temperature using current available experimental techniques. Thermodynamically controlled extrapolation [11] of sublimation pressures measured at higher temperatures therefore seems to be the only method to obtain sublimation thermodynamic data for amino acids at temperatures relevant to their biological activity. For the application of this method, several thermodynamically linked properties are necessary as inputs: (i) sublimation pressure, which was determined at higher temperatures by the Knudsen effusion method for several amino acids including l-proline (from 373 to 416 K) in our previous work [1] (the measurements of other amino acids are in progress); (ii) heat capacity in the ideal gas state, which is typically calculated by the combination of quantum-chemical and statistical thermodynamics methods [3]; and (iii) experimental crystal heat capacities, which were measured in this work for l-histidine, l-phenylalanine, l-proline, l-tryptophan, and l-tyrosine in the temperature range overlapping with that of sublimation pressure measurements by the Knudsen effusion method. Simultaneous correlation of the above-mentioned properties also ensures their thermodynamic consistency.

In order to establish a safe upper temperature limit for the heat capacity measurements, the decomposition temperatures of the amino acids studied were first determined by thermogravimetry. Consequently, after determining the initial crystal structure at 298.15 K by X-ray powder diffraction (XRPD), the phase behavior was studied in the temperature range from 183 K to the decomposition temperature of each amino acid using a heat flux differential scanning calorimeter (DSC) to detect possible phase transitions. Heat capacities in the temperature range from 262 to 358 K were determined with Tian–Calvet calorimetry and extended up to 437 K by using a power compensation DSC. Literature low-temperature heat capacities, which are determined by adiabatic calorimetry from about 10 K to 300 K, were found for all amino acids studied except l-histidine for which new measurements by means of thermal-relaxation calorimetry were performed in this work in the temperature range from 2 to 300 K.

## 2. Results and Discussion

### 2.1. Thermogravimetric Analysis (TGA)

All studied amino acids decompose at/prior to melting using the usual temperature scanning rates (5 to 40) K·min^−1^. The samples were studied by TGA previously and the literature temperatures of decomposition along with the results of this work are summarized in Table 1. It should be noted that the decomposition kinetics is influenced by a number of factors (heating rate, sample size, purge gas, etc.). This is reflected by the relatively large range of reported decomposition temperatures. Despite that, thermograms of this work (shown in Figure 1) agree qualitatively with the literature ones for l-histidine [12,13], l-proline [14], l-phenylalanine [15,16], l-tryptophan [16], and l-tyrosine [16].

The decomposition behavior of the amino acids studied exhibits various patterns. The simplest decomposition path can be observed in the case of l-tyrosine (Figure 1e), which shows a simple one-stage decomposition. For l-histidine (Figure 1a) and l-tryptophan (Figure 1d), two decomposition phases can be clearly distinguished. The situation is more complicated in the case of l-phenylalanine (Figure 1b) and l-proline (Figure 1c). In both cases, four peaks can be observed on the heat flow curve (red), two of which are, however, quite small. In the case of l-proline, the first small peak at 484 K does not seem to be accompanied by any mass loss, pointing to a possible crystal–crystal transition. Note that the temperatures of fusion reported by Do et al. [17] (see Table 1) are higher than those of decomposition by (20 to 100) K. In order to avoid decomposition, Do et al. [17] used scanning rates up to 20,000 K·s ^−1^.

**Table 1 molecules-26-04298-t001:** Temperatures of fusion and decomposition of l-histidine, l-phenylalanine, l-proline, l-tryptophan, and l-tyrosine ^a^.

Reference	*T*_decomp, onset_/K	*T*_decomp, peak top_/K	Method	Scanning Rate, Purge Gas
l-histidine (two-stage decomposition) (*T*_fus_ = 619 ± 7 K ^b^)				
SRC recommendation ^c^	560	-	-	-
Wesolowski and Erecinska [18]	-	523	DTA	5 K min^−1^, air
Wesolowski and Erecinska [18]	-	528	DTG	5 K min^−1^, air
Anandan et al. [19]	-	526	TGA	Not described
Weiss et al. [12]	-	553	TGA	5 K min^−1^, argon
Vraneš et al. [13]	535	-	TGA	10 K min^−1^, argon
Vraneš et al. [13]	-	548.8	DSC	10 K min^−1^, argon
This work	539	-	TGA	5 K min^−1^, argon
l-proline (three-stage decomposition) (*T*_fus_ = 527 ± 7 K ^b^)				
SRC recommendation ^c^	494	-	-	-
Rodante et al. [20]	512	-	TGA	10 K min^−1^, nitrogen
Rodante et al. [14,20]	-	507	DSC	10 K min^−1^, nitrogen
Wesolowski and Erecinska [18]	-	468	DTA/TGA ^d^	5 K min^−1^, air
Wesolowski and Erecinska [18]	-	463	DTG	5 K min^−1^, air
This work	484 ^e^	-	TGA	5 K min^−1^, argon
This work	503	-	TGA	5 K min^−1^, argon
l-phenylalanine (four-stage decomposition) (*T*_fus_ = 579 ± 7 K ^b^)				
SRC recommendation ^c^	556	-	-	-
Rodante et al. [20]	537	-	TGA	10 K min^−1^, nitrogen
Rodante et al. [16,20]	-	531 ^f^	DSC	10 K min^−1^, nitrogen
Rodante et al. [16]	-	549 ^g^	DSC	10 K min^−1^, nitrogen
Rodante et al. [20]	-	544 ^g^	DSC	10 K min^−1^, nitrogen
Lu et al. [21]	-	535 ^g^	DSC	10 K min^−1^, nitrogen
Lu et al. [21]	-	550 ^g^	DSC	10 K min^−1^, nitrogen
This work	511 ^f^	-	TGA	5 K min^−1^, argon
This work	539 ^g^	-	TGA	5 K min^−1^, argon
l-tryptophan (two-stage decomposition) (*T*_fus_ = 620 ± 7 K ^b^)				
SRC recommendation ^c^	564	-	-	-
Rodante et al. [20]	565	-	TGA	10 K min^−1^, nitrogen
Rodante et al. [16,20]	-	570	DSC	10 K min^−1^, nitrogen
This work	554	-	TGA	5 K min^−1^, argon
l-tyrosine (one-stage decomposition) (*T*_fus_ = 678 ± 7 K ^b^)				
SRC recommendation ^c^	616	-	-	-
Rodante et al. [20]	551	-	TGA	10 K min^−1^, nitrogen
Rodante et al. [16,20]	-	591	DSC	10 K min^−1^, nitrogen
Refat et al. [22]	-	596	TGA	10 K min^−1^, nitrogen
This work	579	-	TGA	5 K min^−1^, argon

^a^ Sources where melting/decomposition temperatures are merely mentioned are not listed. Values in this table are rounded to the nearest Kelvin. ^b^ Fast scanning DSC was used by Do et al. [17] for *T*_fus_ determination (see text). ^c^ “Temperature of fusion” (in fact *T*_decomp_) recommended by the Syracuse Research Corporation and used by many authors. ^d^ DTA stands for differential thermal analysis. ^e^ Phase transition assumed (mass loss not detected). ^f^ Small peak preceding the main decomposition peak. ^g^ Main decomposition peak.

### 2.2. Phase Behavior

All amino acids studied are zwitterionic crystals at room temperature. Their initial crystal structures determined at 298.15 K by XRPD are provided in Table 2. Subsequently, the phase behavior was investigated in the temperature range from 183 K to the thermal decomposition temperature using heat-flux DSC for all amino acids studied to confirm/exclude the presence of phase transitions. Except for l-phenylalanine, no phase transitions were detected in the temperature range studied. A subtle endothermic peak on heating was detected for l-phenylalanine at around 440 K (Appendix A), in agreement with Cuppen et al. [23], who, based on a detailed study of this phenomenon by a combination of experimental and computational methods, ascribed this feature to the conformational phase transition of form I (QQQAUJ05) to its high-temperature form Ih. The upper temperature limit of the heat capacity measurements in this work was 437 K in order to observe the trend of the apparent heat capacities in the region of the phase change. The shape of the heat capacity data agrees well with the DSC thermogram (Appendix A), but the peak is, despite its small area, so broad that all the apparent heat capacity data above 422 K had to be discarded. Thus, all the reported heat capacity data for l-phenylalanine corresponds to the form I. For l-proline for which the TGA measurements suggested a phase transition at around 484 K, no visible thermal event was detected in the temperature range from 183 K to its decomposition temperature using heat-flux DSC. Based on the XRPD and DSC analyses along with heat capacity measurements, which did not show any anomalies in the temperature range studied, it can be concluded that the heat capacity data reported in this work corresponds to the crystal structures listed in Table 2.

### 2.3. Heat Capacities

Experimental heat capacities obtained in this work with the SETARAM μDSC IIIa, PerkinElmer DSC 8500, and Quantum Design PPMS are listed in the Appendix A. Available literature data on crystal heat capacities are summarized in Table 3. The heat capacity data for l-phenylalanine, l-proline, and l-tyrosine reported by Cole et al. [30] and those reported for l-tyrosine by Huffman and Ellis [31] are consistent with our measurements and were therefore correlated together with the data listed in Appendix A by Equations (1) and (2), for which the parameters are provided in Table 4. The literature heat capacity data for l-proline [32] were excluded from the correlation because of their slight inconsistency in the low-temperature region with more recent measurements reported in [30].

The experimental heat capacities for all the amino acids studied are compared with the values calculated using Equations (1) and (2) with parameters from Table 4 in Figure 2. The deviations of the selected experimental data from the calculated values do not exceed 2%. Adiabatic heat capacity data by Huffman and Ellis [31] and Cole et al. [30] agree with our measurements within 1%.

The thermodynamic functions derived using Equations (1) and (2) are tabulated in Appendix A and shown in Figure 3.

## 3. Materials and Methods

### 3.1. Samples Description

The title amino acids were of commercial origin and were used as received without further purification. The sample purities (as stated in the certificate of analysis provided by the manufacturer) are reported in Table 5.

### 3.2. Thermogravimetry

Prior to heat capacity and phase behavior measurements, thermogravimetric analysis (TGA) was performed. Traditionally, amino acids are reported to melt or decompose at temperatures above 500 K [5,6]. However, the literature data exhibit a significant scatter. Moreover, a recent work by Weiss et al. [12] suggests that all standard amino acids do not melt but rather decompose, which might damage calorimeters. Weiss et al. [12] reported the evolution of CO_2_, NH_3_, and H_2_O for l-histidine at the decomposition temperature of about 553 K. For the remaining four amino acids of this study, Weiss et al. [12] reported the decomposition without specifying their temperature. Therefore, reinvestigation of the melting/decomposition process for all five amino acids was undertaken to reconcile the literature data. The thermogravimetric analyzer SETARAM Setsys Evolution was used. The samples were placed in an open platinum 100 μL crucible and their thermal stability was investigated in the temperature range from 298 to 573 K with a temperature gradient of 5 K min ^−1^ under an inert Ar atmosphere. 

### 3.3. Phase Behavior Measurements

XRPD was used to characterize initial crystal structures of the amino acids studied using a *θ*–*θ* powder diffractometer X’Pert^3^ Powder from PANalytical in Bragg-Brentano para-focusing geometry using wavelength CuKα radiation (*λ* = 1.5418 Å, *U* = 40 kV, *I* = 30 mA). The samples were scanned at 298.15 K in the range of 5° to 50° 2*θ* with a step size of 0.039° 2*θ* and 0.7 s for each step. The diffractograms were analyzed with the software HighScore Plus in combination with yearly updated powder diffraction databases PDF4+ and PDF4/Organics.

The heat flux DSC TA Q1000 was used for investigation of the phase behavior of the amino acids studied in the temperature range from 183 K to the decomposition temperature. The combined expanded uncertainty (0.95 level of confidence) in the phase transition temperatures and enthalpies is estimated to be 0.3 K and 3%, respectively. 

### 3.4. Heat Capacity Measurements

A Tian–Calvet type calorimeter (SETARAM μDSC IIIa) with operating temperature range 262 K to 358 K was used for the measurement of heat capacities. As the detailed description of the calorimeter and its calibration and operation was reported previously [4], only the most salient information is provided here. The heat capacity measurements were carried out by the continuous heating method [33] using the three-step methodology, i.e., the measurement of the sample is followed by the measurement of the reference material (synthetic sapphire, NIST Standard reference material No. 720) and by performing the blank experiment. The saturated molar heat capacities *C*_sat_ obtained in this work are identical with the isobaric molar heat capacities $C_{pm}^{o}$ in the temperature range studied given the very low sublimation pressures of amino acids. The combined expanded uncertainty (0.95 level of confidence) of the heat capacity measurements is estimated to be $U_{c}\left(C_{pm}^{o}\right)=0.01\cdot C_{pm}^{o}$.

The PerkinElmer power compensation DSC 8500 equipped with autosampler was used for the heat capacity determination in the temperature range from 307 to 437 K. The heat capacity measurements were carried out by the temperature increment method and repeated three times to eliminate systematic errors. The combined expanded uncertainty (0.95 level of confidence) of the heat capacity measurement is estimated to be $U_{c}\left(C_{pm}^{o}\right)=0.03\cdot C_{pm}^{o}$. Due to a lower accuracy, the results obtained by this calorimeter were slightly adjusted to agree with those determined by the more accurate SETARAM μDSC IIIa, following the common practice [34].

For low-temperature heat capacity measurements of l-histidine, a commercially available apparatus Physical Property Measurement System (PPMS) Model 6000 EverCool II (Quantum Design, USA) equipped with a heat capacity module (^4^He, *T*_min_ = 1.8 K) was used. The calorimeter uses the thermal-relaxation measurement technique which is an alternative to the time-intensive and labor-intensive adiabatic calorimetry. The specific heat capacity of a sample is determined by measuring the thermal response to a change in heating conditions [35]. Uncertainty in the heat capacity obtained using PPMS was investigated recently in our laboratory [36]. Based on testing with several compounds [36], the combined expanded uncertainty (0.95 level of confidence) of the heat capacity measurements is estimated to be $U_{c}\left(C_{pm}^{o}\right)=0.10\cdot C_{pm}^{o}$ below 10 K, $U_{c}\left(C_{pm}^{o}\right)=0.03\cdot C_{pm}^{o}$ in the temperature range from 10 to 40 K and $U_{c}\left(C_{pm}^{o}\right)=0.02\cdot C_{pm}^{o}$ in the temperature range from 40 to 300 K.

In order to describe the temperature dependence of heat capacity in a wide temperature range (including literature heat capacities obtained using adiabatic calorimetry and data obtained by Quantum Design PPMS), the following equation proposed by Archer [37] was used: (1)$$C_{pm}^{o}/C_{pm}^{\mathrm{ref}}={\left(\frac{T}{T^{\mathrm{ref}}f(T)+bT}\right)}^{3}$$
where *T*^ref^ = 1 K and $C_{pm}^{\mathrm{ref}}$ = 1 J∙K ^−1^∙mol ^−1^. *f*(*T*) is defined by the following equation:(2)$$f(T)=a_{i}{\left(T-T_{i}\right)}^{3}+b_{i}{\left(T-T_{i}\right)}^{2}+c_{i}\left(T-T_{i}\right)+d_{i}$$
where only a single parameter *d_i_* per each temperature interval is to be optimized, while the values of the other three are imposed by a constraint of continuity and smoothness of the resulting temperature dependence. Parameter *b* is estimated from the slope of *f*(*T*) at temperatures greater than 70 K prior to the optimization procedure [37].

## 4. Conclusions

The decomposition temperatures of the studied amino acids were determined by means of TGA (see Table 2). The thermograms obtained qualitatively agree with those found in the literature. The phase behavior was studied with a heat flux differential scanning calorimetry (calorimeter TA Q1000) in the temperature range from 183 K to the decomposition temperature to determine the presence of possible phase transitions. For l-phenylalanine, a subtle endothermic event corresponding to form I to form Ih transition was detected at around 440 K, in accordance with [23]. New heat capacity values were obtained for l-histidine, l-phenylalanine, l-proline, l-tryptophan, and l-tyrosine (Appendix A). Accurate equations in the temperature range from 0 to 437 K for crystal the heat capacity were established. Equations are based on new heat capacity measurements performed in this work as well as on low-temperature heat capacity data from the literature which showed an excellent agreement with our data in the overlapping temperature interval. Standard thermodynamic functions (entropies, enthalpies, and Gibbs energies) of the crystalline phase at *p* = 0.1 MPa were evaluated in the temperature range from 0 to 430 K (Appendix A).

## Figures and Tables

**Figure 1 molecules-26-04298-f001:**
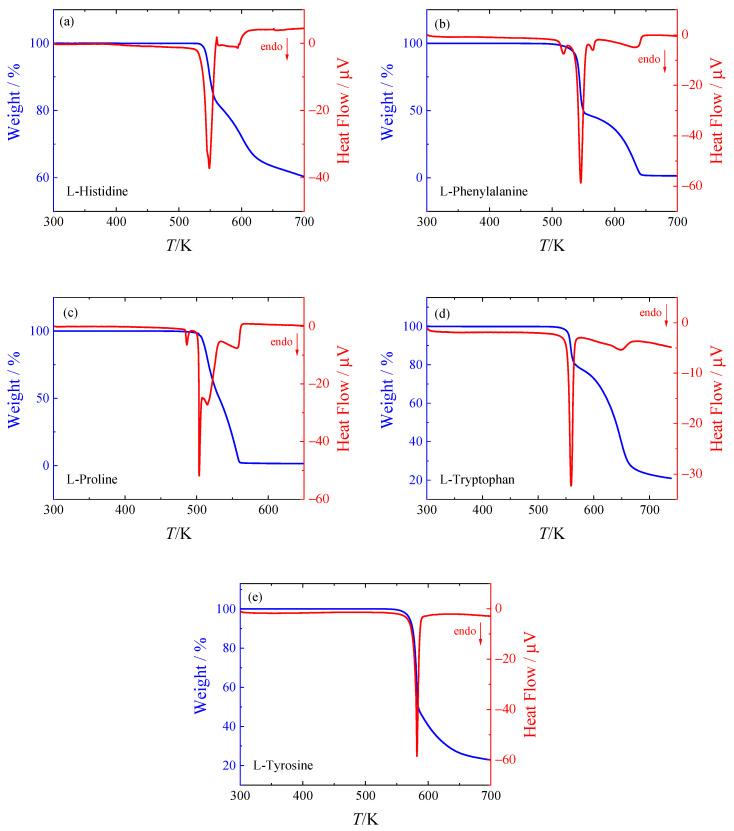
TGA analysis of (**a**) l-histidine, (**b**) l-phenylalanine, (**c**) l-proline, (**d**) l-tryptophan, and (**e**) l-tyrosine.

**Figure 2 molecules-26-04298-f002:**
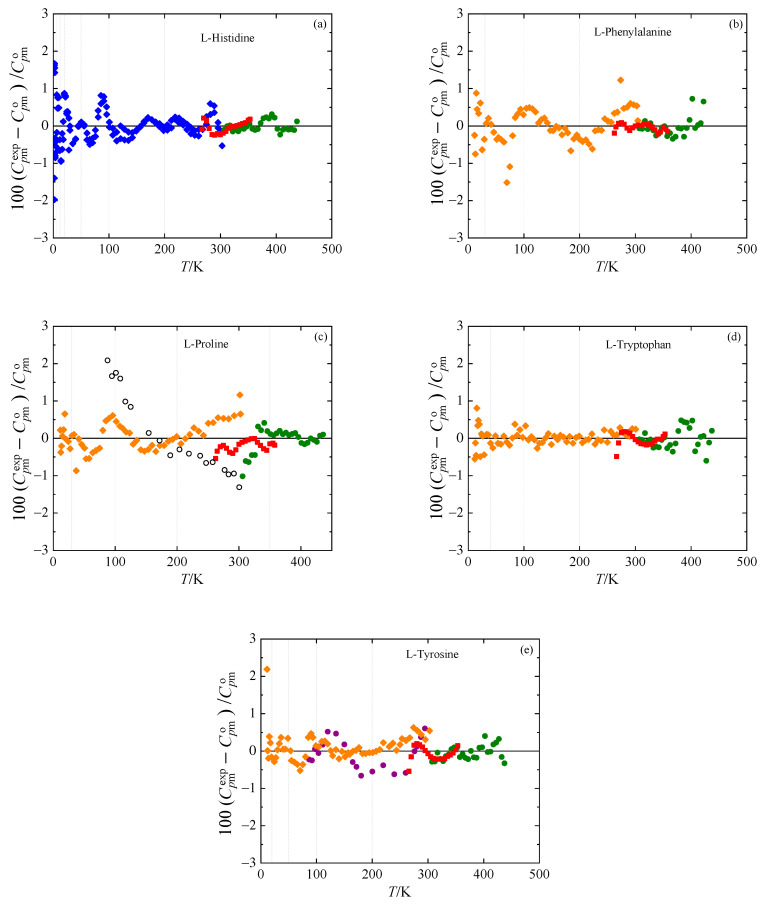
Relative deviations $100\left(C_{pm}^{\exp}-C_{pm}^{o}\right)/C_{pm}^{o}$ of individual experimental heat capacities $C_{pm}^{\exp}$ from values $C_{pm}^{o}$ calculated by means of Equations (1) and (2) with parameters from Table 4. (**a**) l-histidine, (**b**) l-phenylalanine, (**c**) l-proline, (**d**) l-tryptophan, and (**e**) l-tyrosine. Blue diamonds: this work (relaxation calorimetry); red squares: this work (Tian–Calvet calorimetry); green circles: this work (power compensation DSC); orange diamonds: Cole et al. [30]; empty circles: Huffman and Fox [32]; purple circles: Huffman and Ellis [31]. Vertical lines denote the knot temperatures *T_i_*.

**Figure 3 molecules-26-04298-f003:**
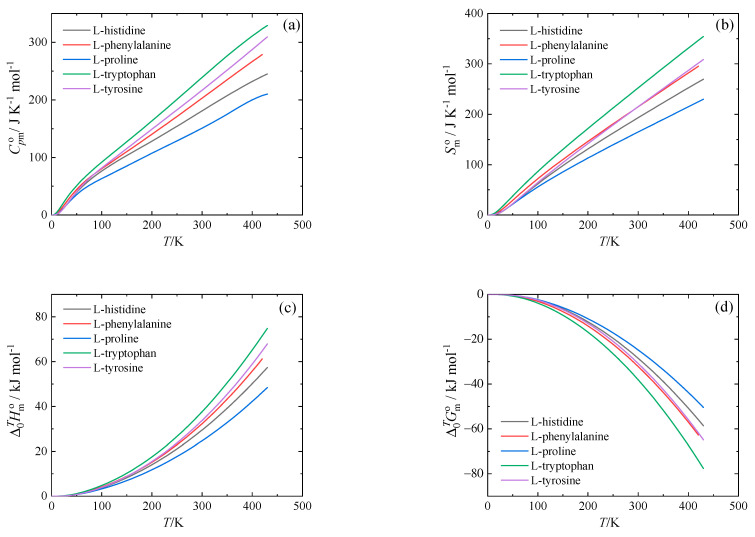
Standard molar thermodynamic functions at *p* = 0.1 MPa. (**a**) Isobaric heat capacity, (**b**) entropy, (**c**) enthalpy, and (**d**) Gibbs energy.

**Table 2 molecules-26-04298-t002:** Initial crystal structures of amino acids studied at (298.15 ± 3) K and *p* = (100 ± 5) kPa.

Compound	Refcode ^a^	*Z*	Space Group	Ref. ^b^
l-histidine	LHISTD10 ^c^	4	*P*2_1_2_1_2_1_	[24]
l-phenylalanine	QQQAUJ05 ^d^	4	*P*2_1_	[25]
l-proline	PROLIN	4	*P*2_1_2_1_2_1_	[26]
l-tryptophan	VIXQOK	16	*P*1	[27]
l-tyrosine	LTYROS11	4	*P*2_1_2_1_2_1_	[28]

^a^ Identifier in the Cambridge Structural Database. ^b^ Reference in which crystal structure parameters with a given Refcode were determined. ^c^ LHISTD10 (conjointly labeled as form A, orthorhombic) is the thermodynamically stable form at room temperature compared to LHISTD01(form B, monoclinic) [29]. ^d^ Conjointly labeled as form I [23].

**Table 3 molecules-26-04298-t003:** Overview of the literature crystal heat capacities of l-histidine, l-phenylalanine, l-proline, l-tryptophan, and l-tyrosine.

Reference	*N* ^a^	(*T*_min_–*T*_max_)/K	*u_r_*(*C_p_*_m_)/% ^b^	Calorimeter
l-histidine				
This work	19	267–353	1.0	Tian–Calvet
This work	126	2–303	^c^	relaxation
This work	27	307–437	3.0	PC DSC ^d^
l-phenylalanine				
Cole et al. [30]	61	11–305	0.2	adiabatic
This work	21	262–358	1.0	Tian–Calvet
This work	24	307–422	3.0	PC DSC
l-proline				
Huffman and Fox [32]	18	88–300	1.0	adiabatic
Cole et al. [30]	55	11–302	0.2	adiabatic
This work	21	262–358	1.0	Tian–Calvet
This work	27	306–436	3.0	PC DSC
l-tryptophan				
Cole et al. [30]	63	12–301	0.2	adiabatic
This work	19	266–353	1.0	Tian–Calvet
This work	27	307–437	3.0	PC DSC
l-tyrosine				
Huffman and Ellis [31]	19	87–295	1.0	adiabatic
Cole et al. [30]	61	11–303	0.2	adiabatic
This work	19	266–353	1.0	Tian–Calvet
This work	27	307–437	3.0	PC DSC

^a^*N* = number of data points. ^b^ *u*_r_(*C_p_*_m_) stands for relative uncertainty in heat capacity as stated by the authors. ^c^ For specification of *u*_r_(*C_p_*_m_) of PPMS using the thermal relaxation measurement technique see Section 3.4. ^d^ PC-DSC stands for power compensation differential scanning calorimetry.

**Table 4 molecules-26-04298-t004:** Parameters of Equations (1) and (2) for crystal heat capacities in J K ^−1^ mol ^−1^.

**l-Histidine *b* = 0.14**							
***a*_i_**	***b*_i_**	***c*_i_**	***d*_i_**	$T_{i}$ **/K**	$T_{i}^{\max}$ **/K**	***N*^a^**	***s*_r_^b^**
−1.15080 × 10^−2^	1.51671 × 10^−1^	−8.09852 × 10^−1^	1.13856 × 10^1^	0	5	12	1.48
1.75300 × 10^−3^	−2.09483 × 10^−2^	−1.56237 × 10^−1^	9.68966 × 10^0^	5	12	10	0.79
−5.49432 × 10^−4^	1.58648 × 10^−2^	−1.91822 × 10^−1^	8.17082 × 10^0^	12	20	8	0.98
−2.92687 × 10^−5^	2.67837 × 10^−3^	−4.34769 × 10^−2^	7.37027 × 10^0^	20	50	16	0.50
−1.12162 × 10^−6^	4.41923 × 10^−5^	3.82001 × 10^−2^	7.68625 × 10^0^	50	100	20	0.48
−9.85627 × 10^−8^	−1.24051 × 10^−4^	3.42071 × 10^−2^	9.56653 × 10^0^	100	200	30	0.20
2.88835 × 10^−7^	−1.53620 × 10^−4^	6.44011 × 10^−3^	1.16482 × 10^1^	200	437	76	0.19
**l-phenylalanine *b* = 0.13**							
***a*_i_**	***b*_i_**	***c*_i_**	***d*_i_**	$T_{i}$ **/K**	$T_{i}^{\max}$ **/K**	***N*^a^**	***s*_r_^b^**
−2.32742 × 10^−5^	2.28357 × 10^−3^	−2.67160 × 10^−2^	5.99138 × 10^0^	0	30	8	0.81
−2.05435 × 10^−6^	1.88887 × 10^−4^	4.74577 × 10^−2^	6.61671 × 10^0^	30	100	15	0.37
3.81465 × 10^−7^	−2.42525 × 10^−4^	4.37031 × 10^−2^	1.01597 × 10^1^	100	200	19	0.34
1.82220 × 10^−7^	−1.28086 × 10^−4^	6.64191 × 10^−3^	1.24862 × 10^1^	200	422	63	0.31
**l-proline *b* = 0.14**							
***a*_i_**	***b*_i_**	***c*_i_**	***d*_i_**	$T_{i}$ **/K**	$T_{i}^{\max}$ **/K**	***N*^a^**	***s*_r_^b^**
−1.33366 × 10^−4^	1.24956 × 10^−2^	−3.51312 × 10^−1^	1.01741 × 10^1^	0	30	9	0.40
−3.72570 × 10^−6^	4.92687 × 10^−4^	3.83360 × 10^−2^	7.27987 × 10^0^	30	100	15	0.44
6.30381 × 10^−7^	−2.89711 × 10^−4^	5.25443 × 10^−2^	1.10996 × 10^1^	100	200	17	0.28
−8.71615 × 10^−8^	−1.00596 × 10^−4^	1.35136 × 10^−2^	1.40873 × 10^1^	200	350	42	0.41
2.46859 × 10^−6^	−1.39819 × 10^−4^	−2.25488 × 10^−2^	1.35568 × 10^1^	350	436	20	0.13
**l-tryptophan *b* = 0.12**							
***a*_i_**	***b*_i_**	***c*_i_**	***d*_i_**	$T_{i}$ **/K**	$T_{i}^{\max}$ **/K**	***N*^a^**	***s*_r_^b^**
−1.06285 × 10^−5^	1.07946 × 10^−3^	3.27392 × 10^−2^	4.46147 × 10^0^	0	40	13	0.49
−1.35546 × 10^−7^	−1.95960 × 10^−4^	6.80791 × 10^−2^	6.81795 × 10^0^	40	100	13	0.21
3.01917 × 10^−7^	−2.20358 × 10^−4^	4.31000 × 10^−2^	1.01680 × 10^1^	100	200	19	0.14
2.55471 × 10^−7^	−1.29783 × 10^−4^	8.08593 × 10^−3^	1.25763 × 10^1^	200	437	64	0.22
**l-tyrosine *b* = 0.13**							
***a*_i_**	***b*_i_**	***c*_i_**	***d*_i_**	$T_{i}$ **/K**	$T_{i}^{\max}$ **/K**	***N*^a^**	***s*_r_^b^**
1.19376 × 10^−4^	−5.84349 × 10^−3^	8.01020 × 10^−2^	7.29354 × 10^0^	0	20	5	0.51
−1.21705 × 10^−5^	1.31909 × 10^−3^	−1.03860 × 10^−2^	7.51319 × 10^0^	20	50	9	0.30
−3.02027 × 10^−6^	2.23746 × 10^−4^	3.58991 × 10^−2^	8.06019 × 10^0^	50	100	14	0.36
4.08232 × 10^−7^	−2.29294 × 10^−4^	3.56217 × 10^−2^	1.00370 × 10^1^	100	200	26	0.27
9.54027 × 10^−8^	−1.06824 × 10^−4^	2.00989 × 10^−3^	1.17144 × 10^1^	200	437	70	0.28

^a^ *N* stands for the number of experimental data points in the given temperature interval used for correlation. ^b^
$s_{r}=100{\left\{\sum_{i=1}^{n}{\left[\left(C_{pm}^{\exp}-C_{pm}^{o}\right)/C_{pm}^{o}\right]}_{i}^{2}/\left(N-m\right)\right\}}^{1/2}$, where $C_{pm}^{\exp}$ and $C_{pm}^{o}$ are the experimental and calculated (by means of Equations (1) and (2)) heat capacity, *N* is the number of data points fitted, and *m* is the number of independent adjustable parameters.

**Table 5 molecules-26-04298-t005:** Sample description.

Compound	CAS Number	Supplier	Mole Fraction Purity ^a^	Purity Method ^a^
l-histidine	71-00-1	Sigma-Aldrich	1.00 ^b^	titration HClO_4_
l-phenylalanine	63-91-2	Fisher Scientific	1.00 ^b^	nosp ^c^
l-proline	147-85-3	Sigma-Aldrich	1.00 ^b^	TLC ^d^
l-tryptophan	73-22-3	Fisher Scientific	1.00 ^b^	nosp ^c^
l-tyrosine	60-18-4	Sigma-Aldrich	>0.99	TLC ^d^

^a^ From the certificate of analysis supplied by the manufacturer. ^b^ Impurities below the detection limit of a given analytical method. ^c^ nosp stands for not specified. ^d^ TLC stands for thin-layer chromatography.

## Data Availability

The data presented in this study are available in the Appendix A.

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
