# Peer review of "Heat Capacities of l-Histidine, l-Phenylalanine, l-Proline, l-Tryptophan and l-Tyrosine"

_molecules, 2021, doi:10.3390/molecules26144298_

Round 1
Reviewer 1 Report
In my opinion this manuscript only needs the correction of the references.
I read the paper very carefully. It's very well written and very clear, I didn't find any jackdaws and I have not any suggestions for improvement. It is very well structured and the results are of great quality.
Author Response
Point 1. In my opinion this manuscript only needs the correction of the references. I read the paper very carefully. It's very well written and very clear, I didn't find any jackdaws and I have not any suggestions for improvement. It is very well structured and the results are of great quality.
Response 1: Thank you for the encouraging review. The references were checked and corrected.
Reviewer 2 Report
see attachment
Heat capacities of L-histidine, L-phenylalanine, L-proline, L-tryptophan, and L-tyrosine
Václav Pokorný, Vojtěch Štejfa, Jakub Havlín, Květoslav Růžička and Michal Fulem
The authors report a careful study of the heat capacity of the title amino acids using a variety of calorimeters over a wide temperature range. The work appears to be carefully performed. I have made a few editorial suggestions but otherwise the narrative is clear and well written. The manuscript is being returned with the suggested changes highlighted.

Author Response
Point 1. The authors report a careful study of the heat capacity of the title amino acids using a variety of calorimeters over a wide temperature range. The work appears to be carefully performed. I have made a few editorial suggestions but otherwise the narrative is clear and well written. The manuscript is being returned with the suggested changes highlighted.
Response 1. Thank you for the encouraging review and editorial suggestions that have been implemented in the revised manuscript.
Reviewer 3 Report
The present paper, with is in line with previous published by the some of the authors, is a useful contribution to the literature and should be published in the present form. It reports very accurate thermophysical experimental results, concretely crystal heat capacities, over a wide range covering the liquid helium temperature up to ca. 430 K, for five amino acids.
The authors performed a characterization of the samples used both in terms of chemical and phase purity using DSC and XRPD techniques. In my opinion a complementary analysis should include chromatography coupled to mass spectrometry. It would serve as a validation (double check) of the purity indicated by the manufacturer.
Although some temperature ranges for these compounds are already published in the literature basically using adiabatic calorimetry, the authors have extended the range and this is clearly shown in Table 4. For L-histidine the experimental measurements were more exhaustive and the low temperature range was covered using relaxation (heat pulse) calorimetry. This method being (usually) less rigorous than adiabatic calorimetry has the advantage of being less time consuming.
Finally, the authors, using equations 1 and 2, describe the dependence of the heat capacity on temperature for various temperature ranges and the curves of fit are presented in Fig. 3.
Some items to consider in the final version of the manuscript:
Pag 4 (section 3.1) – Instead of “the situation is more complicated in the case of L-phenylalanine (Figure 1c) and L-proline (Figure 1d).”, please consider “the situation is more complicated in the case of L-phenylalanine (Figure 1b) and L-proline (Figure 1c).”
Pag 5 – Fig. 1 – in my opinion an enlargement of the heat flow line in figure b (insert) should be introduced in order to highlight the thermal event occurring at 440 K.
Pag 10 – As a suggestion: for a potential reader unfamiliar with the relationship between thermodynamic quantities the indication of the equations used to represent the results shown in figures 3(b) to (d) should be placed in the text, or alternatively in the supplementary materials.
Author Response
Point 1. The present paper, with is in line with previous published by the some of the authors, is a useful contribution to the literature and should be published in the present form. It reports very accurate thermophysical experimental results, concretely crystal heat capacities, over a wide range covering the liquid helium temperature up to ca. 430 K, for five amino acids.
The authors performed a characterization of the samples used both in terms of chemical and phase purity using DSC and XRPD techniques. In my opinion a complementary analysis should include chromatography coupled to mass spectrometry. It would serve as a validation (double check) of the purity indicated by the manufacturer.
Although some temperature ranges for these compounds are already published in the literature basically using adiabatic calorimetry, the authors have extended the range and this is clearly shown in Table 4. For L-histidine the experimental measurements were more exhaustive and the low temperature range was covered using relaxation (heat pulse) calorimetry. This method being (usually) less rigorous than adiabatic calorimetry has the advantage of being less time consuming.
Finally, the authors, using equations 1 and 2, describe the dependence of the heat capacity on temperature for various temperature ranges and the curves of fit are presented in Fig. 3.
Response 1. Thank you for the encouraging review. Based on the discussion with our colleagues from the Central laboratories of UCT Prague, who operate the gas-liquid chromatograph coupled with mass spectrometer, this technique is not suitable suitable for analyzing the purity of amino acids.
Point 2. Pag 4 (section 3.1) – Instead of “the situation is more complicated in the case of L-phenylalanine (Figure 1c) and L-proline (Figure 1d).”, please consider “the situation is more complicated in the case of L-phenylalanine (Figure 1b) and L-proline (Figure 1c).”
Response 2. Corrected. Thank you.
Point 3. Pag 5 – Fig. 1 – in my opinion an enlargement of the heat flow line in figure b (insert) should be introduced in order to highlight the thermal event occurring at 440 K.
Response 3. Please refer to Figure S1 in the Supplementary Materials which shows the phase transition of L-phenylalanine occurring at 440 K in detail.
Point 4. Pag 10 – As a suggestion: for a potential reader unfamiliar with the relationship between thermodynamic quantities the indication of the equations used to represent the results shown in figures 3(b) to (d) should be placed in the text, or alternatively in the supplementary materials.
Response 4. The thermodynamic relationships used to calculate thermodynamic functions using heat capacity data were added to the Supplementary Materials (a short introduction to the section "Tabulated thermodynamic functions" was added).